# Developing a typology of models of palliative care delivery in prisons in high-income countries: protocol for a scoping review with narrative synthesis

Emma Gilbert ![ORCID],[1] M Turner,[2] Nick de Viggiani,[3] Lucy Selman[4]

¹Palliative and End of Life Care Research Group, Bristol Medical School, University of Bristol, Bristol, UK
²School of Human and Health Sciences, University of Huddersfield, Huddersfield, UK
³School of Health and Social Wellbeing, University of the West of England Bristol, Bristol, UK
⁴Palliative and End of Life Care Research Group, Bristol Medical School, University of Bristol Faculty of Health Sciences, Bristol, UK

**Correspondence to**
Emma Gilbert;
emma.gilbert@bristol.ac.uk

## ABSTRACT

**Introduction** A combination of punitive sentencing practices within ageing populations, compounded by the health challenges faced by people in prison, means that dedicated palliative care provision within prisons is a pressing requirement. However, evidence about exactly how quality palliative and end-of-life care is delivered in this environment remains sparse.

This review aims to develop a typology of models of palliative and end-of-life care delivery within prisons in high-income countries to inform service development and policy.

**Methods and analysis** We will conduct a scoping review of published studies and grey literature, following the Arksey and O'Malley framework. We will report data on models of palliative and end-of-life care delivery in prisons in high-income countries. Searches will be undertaken in Medline, EMBASE, CINAHL, Social Sciences Citation Index and PsyINFO for all study types, published from 1 January 2000 to December 2021, and reference lists from key reviews and studies will be screened for additional references. We will also screen grey literature from within other high-income countries using a targeted search strategy. For published reports of original research, study quality and risk of bias will be assessed independently by two reviewers using the Mixed Methods Appraisal Tool. A narrative synthesis of the data will be undertaken, integrating the results of the quality assessment.

**Ethics and dissemination** Approval by research ethics committee is not required since the review only includes published and publicly accessible data. We will publish our findings in a peer-reviewed journal as per Preferred Reporting Items for Systematic Reviews and Meta-Analyses 2020 guidance.

**Protocol registration** The final protocol was registered with the Research Registry on 26 November 2021 (www.researchregistry.com).
Unique ID number: reviewregistry1260.

## Strengths and limitations of this study

► This protocol conforms to the Preferred Reporting Items for Systematic Reviews and Meta-Analyses 2020 guidelines.
► The interpretation of 'models of care' escapes clear definition within the research literature, so it is not possible to include it in the search strategy; this information will be extracted and a typology developed using a prepiloted data extraction template.
► We adopt a narrative synthesis approach as initial searches suggest that the studies identified will be insufficiently similar in research design and there will be a high volume of grey literature such as policy documents and statutory reports.
► Narrative synthesis will provide an in-depth understanding of the literature on how palliative care is delivered in prisons across high-income countries, informing subsequent research.

## INTRODUCTION

This review intends to develop a typology of models of palliative care delivery within prisons in high-income countries. A combination of punitive sentencing practices within ageing populations, compounded by the health challenges faced by people in prison,

means that dedicated palliative care provision is a pressing requirement within many prisons.[1 2] However, evidence about exactly how quality end-of-life care is delivered in these environments remains sparse.[1]

With the largest prison population in Western Europe, the demographic of older people in prison is growing rapidly within England and Wales (prisons are devolved within the UK). People aged 60 years and over are the fastest growing age group in the prison estate, which is three times as many as 16 years ago.[3] This trend is visible across Europe—of the 48 prison administrations providing data for the Council of Europe's 2020 SPACE report on prison indicators, 14.8% of inmates were aged 50 years or over. Imprisoned individuals living behind bars now represent the fastest growing group in correctional facilities in the UK, as well as Australia, Switzerland, Japan and the USA.[4] Table 1 illustrates the percentage increase of older people in prison across some

**Table 1** Percentage increase of older age people in prison across high-income countries

| Date range | Country | Age | Percentage increase | Source of data |
|---|---|---|---|---|
| 2013–2018 | Singapore | 60+ | 50 | 23 |
| 2013–2018 | South Korea | 65+ | 45 | 23 |
| 2007–2017 | Switzerland | 50+ | 100 | 23 |
| 1990–2030 | USA | 55+ | 4400 | 23 |
| 2010–2019 | Canada | 50+ | 50 | 23 |
| 2002–2020 | UK | 60+ | 243 | 24 |
| 2000–2010 | Australia | 65+ | 84 | 25 |
| 2000–2009 | New Zealand | 50+ | 94 | 25 |

high-income countries; the numbers are expected to increase significantly in coming years.[5]

There is evidence of an association between incarceration and poor health outcomes.[6] Prisons tend to accumulate individuals who have experienced significant health inequalities, with far greater incidences of mental health and substance misuse disorders, as well as physical health comorbidities, than the general population.[2] These health disparities are often intensified by the environmental challenges of delivering healthcare within the built environment. Ageing buildings which cannot ensure rigorous infection prevention control; cells that lack adequate space for specialist equipment; and a regime that imposes limitations on an individual's self-efficacy regarding their own nutrition, physical activity, relaxation and sleep inevitably affect an individual's ability to cope.[7] People in prison consequently face increased morbidity.[8] In a 2018 rapid review, the estimated annual prevalence of those requiring end-of-life care in French prisons was twice as high as the anticipated equivalent expected in the general population, and comparable with a population 10 years older.[8]

Research into palliative care within the penal system is an emerging area, and substantial gaps remain regarding the current nature of provision and best practice models. Recent investigation by the European Association for Palliative Care Task Force for Prisoners addressed some of these through data collection within eight countries, examining palliative care provision, causes of death in custody and the application of early release on compassionate grounds policies.[1] This research highlighted the inequitable provision for those either dying or living with a life-limiting illness in prison, as well as the limited potential that current early release policies offer in practice.[1] Other salient research has focused on the ethical challenges that delivering palliative care within a human rights framework poses within the prison system,[9] the experience of terminal illness while incarcerated,[4] as well as the 'double burden' experienced by older people in prison who face additional suffering from the failure of prison healthcare to adequately meet their needs.[10]

In the UK, understanding the palliative and end-of-life care needs for people in prison has gained traction and many prisons have well-coordinated relationships with their local palliative care teams and hospices.[11] The publication of the Dying Well in Custody Charter–End of Life Care Ambitions[12] articulated these developments as a set of standards for end-of-life care in prisons, but there is variation in how the charter has been applied.

Findings from this review will help ensure that best available evidence informs future provision of culturally relevant, tailored palliative and end-of-life care and support for people in prison. The evidence from this review will also provide a basis for policymaking for health and correctional service procedure and protocol around early release on compassionate grounds and alternative secure accommodation for ageing people in prison and those experiencing life-limiting illness.

In accordance with guidelines, our scoping review protocol was registered with the Research Registry on 26 November 2021 (ID: reviewregistry1260).

## AIM

This scoping review aims to map and synthesise the literature on models of palliative and end-of-life care for people in prison, within prisons in high-income countries. Its objectives are to describe models of service delivery that currently exist in published and grey literature, appraise these models in terms of outcomes and impact, and describe facilitators of and the challenges in delivering different models of palliative and end-of-life care for people in prison. The synthesis will then consider how the identified models meet the overall intentions of palliative care as defined by the WHO,[13] drawing out implications and recommendations for service provision and policy.

## REVIEW QUESTIONS
### SPICE framework

Setting–adult prisons, both male and female.

Perspective–prison staff, prison volunteers, patients and their family/carers (who have current or prior experience of a family member receiving end-of-life care in prison).

Intervention–model of palliative care/end-of-life healthcare delivery—be it a prison hospice, specialist

in-reach palliative care provision to a prison or another integrated model.

Comparison—qualitative and mixed-methods studies are unlikely to have a comparison group; quantitative studies may compare the intervention with usual care or with a comparison/control group.

Evaluation—mapping and describing available models of palliative healthcare delivery in terms of acceptability and usefulness to patients, family/carers and clinicians; outcomes and impact of these models; and facilitators of their implementation.

### Primary question

What models of palliative and end-of-life care for people in prison are described in both the published and grey literature?

### Secondary questions

What evidence exists regarding the outcomes and impact of these models?

What are the facilitators of and challenges in delivering different models of palliative and end-of-life care for people in prison?

## METHODS
### Inclusion criteria

Study reports will be included in this scoping review if they meet the following inclusion criteria:

1. Any study reporting new empirical data, regardless of study design.
2. Studies reporting models and mechanisms of palliative and end-of-life healthcare delivery to the prison population within the UK and other comparable high-income countries.
3. Studies reporting the views and experiences of different models of palliative and end-of-life healthcare delivery in prison from the perspective of:
   a. People in prison, their families and informal carers (including in bereavement).
   b. Prison staff and volunteers.
4. Studies conducted in high-income countries that are published in English. High-income countries are defined by the World Bank as having a gross national income per capita exceeding $12 056.[14]
5. Studies reported since 1 January 2000 until 11 December 2021.

### Exclusion criteria

1. Studies not reported in English.
2. Studies reporting on chronic or life-limiting illness, death and dying within prison and criminal justice contexts where the model of care delivery is not described.
3. Studies about institutions that do not fall under the legal definition of prison (eg, Immigration Removal Centres), or do not cater for adult people in prison (eg, Secure Children's Homes).

4. Studies about prison palliative care, where patient/caregiver or staff experiences are reported, but the model of care delivery is not described or evaluated.
5. Studies that focus on components of palliative care provided at specific phases of the disease trajectory and do not describe the overall model of palliative care delivery (eg, pain management only).
6. Studies from low-income and middle-income countries.
7. Studies published prior to 1 January 2000.

Grey literature such as conference abstracts, audits, theses and dissertations, research and committee reports, government reports, policy documents, quality improvement reports and ongoing research will be included if they present relevant empirical data. If there is uncertainty about whether the inclusion criteria are met, or if relevant data cannot be extracted, the authors will be contacted to ask if they can provide additional information and/or further data. If this is not possible, the study will be excluded.

Adopting the five stages of the Arksey and O'Malley framework as shown in table 2,[15] this review aims to identify all relevant literature available on the topic, regardless of study design. This method is especially advantageous for assembling emerging evidence, as well as being suitable for addressing questions that go beyond the scope of effectiveness of an intervention.[16] The approach adopts an iterative process of study selection, data collation, synthesis and presentation.[17]

This review will build on the five stages of the Arksey and O'Malley framework by including critical appraisal of the quality of published studies, using the Mixed Methods Appraisal Tool (MMAT), V.2018.[18]

### Search strategy

The following databases will be searched for English-language studies:

Medline and EMBASE in Ovid, CINAHL, the Social Sciences Citation Index and PsyINFO.

Additional hand searches of key journals, screening of reference lists of included studies, citation tracking and input from expert collaborators will supplement the database searches. A further exploration of the grey literature will be conducted through searches of key websites (eg, International Association for Hospice and Palliative Care, the WHO) and key grey literature databases (Google Scholar, ProQuest). Forward searches of included articles will be undertaken in Google Scholar to identify recently cited articles to supplement those identified in database searches.

The Medline search strategy is shown in online supplemental material 1. This strategy will be adapted to the other electronic databases and is available to view in online supplemental material 2. Any modifications will be reported in the review manuscript. Database searches were run in December 2021. The expected end date for the review is in September 2022.

**Table 2** Five stages of the Arksey and O'Malley scoping review

| Stage of review | Illustration of decisions and issues |
| --- | --- |
| Identifying the research question | Theoretical and empirical work describing models of palliative and end-of-life care delivery to people in prison in the UK and other high-income countries, with broadly comparable criminal justice systems and approaches to human rights. Greater understanding of best practice and challenges and barriers to access. |
| Identifying relevant studies | Specific search criteria designed with key terms used included palliative care, hospice, end of life, compassionate release, prison, penitentiary, imprisonment, incarceration, jail, custody, advance care planning. |
| Study selection | Final included studies may include a diverse representation of primary sources; data will be extracted using the Joanna Briggs Institute Mixed Methods Data Extraction Form following a Convergent Integrated Approach. |
| Charting the data | Data will be extracted from primary sources, different models of care summarised, best practice and barriers to access identified in a narrative synthesis. |
| Collating, summarising and reporting the results | Recommended models proposed with areas for further research and development identified. |

### Screening and data extraction

Search results from each database will be downloaded and managed in Covidence, an online review management platform.[19]

Each title and abstract will be screened against the inclusion/exclusion criteria by one of the review team members. A second reviewer will independently screen a sample of 25% of the titles and abstracts. Full text will be reviewed if inclusion is unclear based on title and abstract. Any discrepancies of study inclusion will be adjudicated by a third reviewer. Grey literature will be screened and synthesised separately and will not be subject to the same method of quality appraisal.

EG will extract data using a prepiloted, customised data extraction form based on the JBI Mixed Methods Data Extraction Form following a Convergent Integrated Approach,[20] in Covidence. Data extraction will be reviewed by a second reviewer and modified where needed. Discrepancies regarding data extraction will be resolved by discussion and consensus, and if necessary, include a third reviewer.

### Quality assessment

The quality of all included studies published in peer-reviewed journals will be assessed independently by two reviewers using the MMAT, V.2018.[18] This validated tool is appropriate for this review as it can be applied to qualitative, quantitative (randomised, non-randomised and descriptive) and mixed-methods study designs. The tool uses a set of questions specific to study design, converted into four possible scores (worst to best: 25/50/75/100). Disagreements between the reviewers will be resolved through discussion, involving a third reviewer if necessary. No studies will be excluded based on their quality, but the narrative synthesis will reflect on the quality of the identified studies. Grey literature will not be subject to quality appraisal and will be analysed and reported separately.

### Evidence synthesis

A narrative synthesis will be conducted to synthesise the findings of the different studies. Due to the potential range of studies that may be included in this integrative review, a narrative synthesis is the most appropriate way to synthesise the findings.

This review will follow the narrative synthesis approach outlined by Popay *et al.*[21] This process will involve developing a preliminary synthesis, exploring relationships within and between the studies, and assessing the robustness of the synthesis overall.[16]

Grey literature will be synthesised and described where it adds relevant data to the research topic. The narrative synthesis will move beyond simply summarising the main features of included studies, presenting the data in such a way that these enable investigations into similarities and differences between studies, while assessing the data and strength of the evidence.[17] The synthesis will be structured around the core models of palliative and end-of-life care delivery for people in prison. For each model, the following data will be synthesised: effectiveness and impact, facilitators for implementation, challenges and barriers of implementation. Implications for future service delivery, policy and research will be identified.

### Patient and public involvement

We plan to include two members of the public with experience of end-of-life care in prisons in the review, inviting them to comment on the narrative synthesis and resulting implications as coauthors on the published review.

### ETHICS AND DISSEMINATION

This scoping review of published/publicly available studies is exempt from ethical approval. The review will be reported as per Preferred Reporting Items for Systematic Reviews and Meta-Analyses guidance,[22] and published in a peer-reviewed journal.

**Acknowledgements** The authors would like to thank Sarah Herring, University of Bristol Subject Librarian, for support in developing the database searches and Isla Kuhn, Head of Medical Library Services, University of Cambridge Medical Library, for support in developing the grey literature search strategy.

**Contributors** EG initiated and designed the review as part of an NIHR predoctoral fellowship supervised by LS and NdV. LS, MT and NdV contributed to the design of the protocol. EG drafted the manuscript. All the authors contributed to the revision of the manuscript and approved the final version.

**Funding** This work is undertaken as part of EG's National Institute for Health Research (NIHR) predoctoral fellowship (award number NIHR301173). LS is funded by an NIHR Career Development Fellowship.

**Disclaimer** The views expressed in this publication are those of the authors and not necessarily those of the NHS or the NIHR.

**Competing interests** None declared.

**Patient and public involvement** Patients and/or the public were not involved in the design, or conduct, or reporting, or dissemination plans of this research.

**Patient consent for publication** Not required.

**Provenance and peer review** Not commissioned; externally peer reviewed.

**ORCID iD**
Emma Gilbert http://orcid.org/0000-0002-5140-2507

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
