## [Reviewer comments · BMJ Open]

ARTICLE DETAILS

TITLE (PROVISIONAL)	Developing a typology of models of palliative care delivery in prisons in high income countries: protocol for a scoping review with narrative synthesis
AUTHORS	GILBERT, Emma; Turner, M; de Viggiani, Nick; Selman, Lucy

VERSION 1 – REVIEW

REVIEWER	Panozzo, Stacey St Vincent's Hospital Melbourne Pty Ltd, Palliative Nexus I currently undertake research in this area and I am a member of the European Association of Palliative Care Taskforce for Prisoners. I declare that I have published previously with one of the co-authors of this manuscript (MT).
REVIEW RETURNED	14-Feb-2022

GENERAL COMMENTS	Thank you for the opportunity to review this manuscript. This is an important and much needed study to provide evidence and understanding of how palliative and end of life care is delivered within the prison setting across developed countries. Looking forward to reading the outcome of this review in the future. This manuscript is ready for publication pending minor amendments as noted in the comments below. • Paragraph 1 p.9 final sentence beginning “Other salient research has focused on...” is quite lengthy. You highlight some important and impactful points for the reader – perhaps consider breaking this down to 2 or 3 sentences.• Paragraph 2 p.9 add full stop at the end of first sentence• Final paragraph p.9 the last sentence feels a little rushed in articulating the significant ways in which your work will extend on previous research and publications. It would be great to break this down, for example: Findings from this review will help to ensure that best available evidence informs future provision of culturally relevant, tailored palliative and end of life care and support for people in prison. The evidence from this review will also provide a basis for policy making for health and correctional service procedure and protocol around early release on compassionate grounds and alternative secure accommodation for ageing people in
--

	prison and those experiencing life limiting illness.  For language use throughout the article: The use of person-first humanizing language when discussing people in prison is preferable to the use of the term prisoner(s). A previous reviewer once pointed this out to myself and other coauthors, with the following article providing suitable person-first alternative(s) to the term 'prisoner' - see Tran et al 2018: "Words matter: a call for humanizing and respectful language to describe people who experience incarceration". AIM / METHOD  Overall intentions of palliative care – what will be used to guide this? WHO definition or alternative, etc? It would be good in the method section to articulate this more clearly. Inclusion criteria #2 p.11 The inclusion and exclusion criteria are clearly reported. However, reader may need clarity on the word other ("...within other high-income countries..."). Is the United Kingdom not included in this review? Paragraph 1 p.14 Perhaps the end date of the review should (also) be mentioned in the inclusion criteria, just to make clear the full period of inclusion for the review.
--	--

REVIEWER	Loeb, Susan J. Penn State, College of Nursing
REVIEW RETURNED	27-Feb-2022

GENERAL COMMENTS	Thank you for the opportunity to review this thoughtfully designed scoping review protocol and well-written manuscript which describes it. My comments are few:  1. Replace "Psych Info" with "PsyINFO" 2. Please note if forward searches of articles included were performed in Google Scholar or other in order to identify more recent articles citing the literature that you identified via the library data bases searched. 3. When writing end of life, if used as an adjective, you should hyphenate; however, when used as a noun you should not hyphenate.
--

VERSION 1 – AUTHOR RESPONSE

Reviewer 1

Paragraph 1 p.9 final sentence beginning "Other salient research	Thank you for this. We have now condensed this sentence without losing any of the key reference. The sentence appears in the last paragraph on
--	--

has focused on..." is quite lengthy. You highlight some important and impactful points for the reader – perhaps consider breaking this down to 2 or 3 sentences.	p5 – p6.
Paragraph 2 p.9 add full stop at the end of first sentence	Thank you for identifying this, this has now been added.
Final paragraph p.9 the last sentence feels a little rushed in articulating the significant ways in which your work will extend on previous research and publications. It would be great to break this down, for example: Findings from this review will help to ensure that best available evidence informs future provision of culturally relevant, tailored palliative and end of life care and support for people in prison. The evidence from this review will also provide a basis for policy making for health and correctional service procedure and protocol around early release on compassionate grounds and alternative secure accommodation for ageing people in prison and those experiencing life limiting illness.	We really appreciate the reviewer taking the time to articulate the ways in which the review will extend upon current research and have utilised the addition gratefully. It appears on paragraph 3 of p 6. We believe this has enhanced the paper so thanks again.
For language use throughout the article: The use of person-first humanizing language when discussing people in prison is preferable to the use of the term prisoner(s). A previous reviewer once pointed this out to myself and other coauthors, with the following article providing suitable person-first alternative(s) to the term 'prisoner' - see Tran et al	Thank you for highlighting this important issue. Having reviewed the suggested paper we decided on the term 'people in prison' instead of 'prisoners' and have amended throughout the article.

2018: “Words matter: respectful language call for humanizing and to describe people who experience incarceration”.	
AIM/METHOD Overall intentions of palliative care – what will be used to guide this? WHO definition or alternative, etc? It would be good in the method section to articulate this more clearly.	In the AIM section, final paragraph p6, we have made explicit that ‘the synthesis will then consider how the identified models meet the overall intentions of palliative care as defined by the World Health Organization (25), drawing out implications and recommendations for service provision and policy’.
Inclusion criteria #2 p.11 The inclusion and exclusion criteria are clearly reported. However, reader may need clarity on the word other (“...within other high-income countries...”). Is the United Kingdom not included in this review?	Thank you for highlighting this. We have amended the inclusion criteria to ensure readers understand that the United Kingdom is included within the review, and we are aiming to include: ‘Studies reporting models and mechanisms of palliative and end of life healthcare delivery to the prison population within the UK and other comparable high-income countries’.
Paragraph 1 p.14 Perhaps the end date of the review should (also) be mentioned in the inclusion criteria, just to make clear the full period of inclusion for the review.	The database searches were run on the 11th December 2021 with an anticipated end date of 30th September 2022. I have decided not to include specific dates within the published protocol as lead researcher and author, I am just starting maternity leave so there is an element of the unknown regarding timeframes for completion. Hope including months rather than dates is adequate.

Reviewer 2

Replace "Psych Info" with "PsyINFO"	Thank you for highlighting this error, this has now been amended throughout the paper.
Please note if forward searches of articles included were performed in Google Scholar or other in order to identify more recent articles citing the literature that you identified via the library data bases searched.	We have added some information on this, final paragraph p 10 ‘Forward searches of included articles will be undertaken in Google Scholar to identify recent citing articles to supplement those identified in database searches.
When writing end of life, if used as an adjective, you should hyphenate; however, when used as a noun you should not hyphenate	Thank you – this has been amended throughout the paper accordingly.

Editor’s comments

Please include, as a supplementary file, the precise, full search strategy (or strategies) for all databases, registers and websites, including any filters and limits used.	This has now been included as a supplementary file.
In your protocol you state: “The expected end date for the review is 30th April 2021”. Please double check this date as it contradicts the previous sentence: “Databases searches were run on 13th October 2021”, as well as the dates provided in the abstract.	Thanks for highlighting this – the dates have now been amended as the timeframes for completion etc have changed and the searches were re-run. The first paragraph of p11 now states ‘. Databases searches were run in December 2021. The expected end date for the review is in September 2022.’
Please complete a thorough proofread of the text and correct any spelling and grammar errors that you identify.	Thank you – this has now been completed.
Formatting Amendments (where applicable): Please ensure that all affiliations of all the authors are listed in the title page.	Thank you – I have amended on the paper as well as within the author dashboard.